# Design and rationale of the Cardiovascular Health and Text Messaging (CHAT) Study and the CHAT-Diabetes Mellitus (CHAT-DM) Study: two randomised controlled trials of text messaging to improve secondary prevention for coronary heart disease and diabetes

Xiqian Huo,[1] Erica S Spatz,[2] Qinglan Ding,[2] Paul Horak,[2] Xin Zheng,[1] Claire Masters,[2] Haibo Zhang,[1] Melinda L Irwin,[2] Xiaofang Yan,[1] Wenchi Guan,[1] Jing Li,[1] Xi Li,[1] John A Spertus,[3] Frederick A Masoudi,[4] Harlan M Krumholz,[2] Lixin Jiang[1]

XH and ESS contributed equally.

For numbered affiliations see end of article.

**Correspondence to**
Dr Lixin Jiang;
jiangl@fwoxford.org

## ABSTRACT

**Introduction** Mobile health interventions have the potential to promote risk factor management and lifestyle modification, and are a particularly attractive approach for scaling across healthcare systems with limited resources. We are conducting two randomised trials to evaluate the efficacy of text message-based health messages in improving secondary coronary heart disease (CHD) prevention among patients with or without diabetes.

**Methods and analysis** The Cardiovascular Health And Text Messaging (CHAT) Study and the CHAT-Diabetes Mellitus (CHAT-DM) Study are multicentre, single-blind, randomised controlled trials of text messaging versus standard treatment with 6 months of follow-up conducted in 37 hospitals throughout 17 provinces in China. The intervention group receives six text messages per week which target blood pressure control, medication adherence, physical activity, smoking cessation (when appropriate), glucose monitoring and lifestyle recommendations including diet (in CHAT-DM). The text messages were developed based on behavioural change techniques, using models such as the information-motivation-behavioural skills model, goal setting and provision of social support. A total sample size of 800 patients would be adequate for CHAT Study and sample size of 500 patients would be adequate for the CHAT-DM Study. In CHAT, the primary outcome is the change in systolic blood pressure (SBP) at 6 months. Secondary outcomes include a change in proportion of patients achieving a SBP <140 mm Hg, low-density lipoprotein cholesterol (LDL-C), physical activity, medication adherence, body mass index (BMI) and smoking cessation. In CHAT-DM, the primary outcome is the change in glycaemic haemoglobin (HbA$_{1c}$) at 6 months. Secondary outcomes include a change in the proportion of patients achieving HbA$_{1c}$<7%, fasting blood glucose, SBP, LDL-C, BMI, physical activity and medication adherence.

**Ethics and dissemination** The central ethics committee at the China National Center for

### Strengths and limitations of this study

- ► The trials are the first to investigate the efficacy of simple and cost-effective text message-based interventions to support management of coronary heart disease (CHD) and diabetes mellitus (DM) in a large and diverse population in China, and have the potential for scaling across healthcare systems in resource-constrained settings.
- ► The study addressed the role of text messages in managing multiple risk factors in patients with CHD or DM, and also targeted high-risk patients with CHD with or without diabetes.
- ► The Cardiovascular Health and Text Messaging (CHAT) Study and CHAT-DM study are distinguished by their relative large sample sizes and culturally appropriate, theory-driven text messages, which were developed using a behaviour change technique and tailored to a specific Chinese patient population.
- ► In contrast to most prior single-centre text messaging trials, these two studies are conducted at 37 participating sites spread across a large and geographically diverse country. The results of the studies may be more generalisable.
- ► However, the text messages, though semipersonalised (participant's preferred name, smoking status) were not tailored specifically to every individual.

Cardiovascular Disease and the Yale University Institutional Review Board approved the CHAT and CHAT-DM studies. Results will be disseminated via usual scientific forums including peer-reviewed publications.
**Trial registration number** CHAT (NCT02888769) and CHAT-DM (NCT02883842); Pre-results.

## INTRODUCTION

The benefits of secondary prevention strategies for coronary heart disease (CHD) targeting lifestyle modification and risk factor management are well established worldwide,[1] [2] however adoption of these strategies is suboptimal.[3] Smoking, inactivity and obesity are prevalent among people with established CHD and control of hypertension and diabetes are often suboptimal. Additionally, medication adherence is poor. Prior studies revealed that only three-fourths of all hospitalised patients take all medications from their discharge prescriptions by 120 days after discharge.[4] Furthermore, less than half of patients hospitalised with acute myocardial infarction (AMI) are adherent to evidence-based medications 1 year later, with the greatest gaps in adherence occurring in the first 6 months after treatment initiation.[5–7]

In lower-income and middle-income countries (LMICs), including China, which face a growing burden of cardiovascular disease and greater challenges to medication access for secondary prevention, over two-thirds of patients with CHD take no medication.[8–10] While high medication costs are a barrier,[11] there is also limited time for education and consultation regarding lifestyle and medication management during clinic visits, which tend to be very brief.[12] [13] Therefore, innovative and cost-effective interventions to enhance adherence are urgently needed.

Mobile phones are pervasive and thus can be used to deliver interventions that help people adopt secondary prevention strategies for CHD in LMICs. They are a primary, inexpensive and quick form of communication, and are widely used to schedule alerts and reminders. In addition, the number of mobile phone users has grown exponentially in the past decade worldwide, and is projected to reach 4.77 billion by 2017.[14] As of August 2016, China had the largest number of mobile phone owners in the world, at 1.3 billion.[15] Mobile phones are also used across all geographical regions and income levels. Due to their ubiquity and convenience, nearly 200 000 mobile phone messages are sent every second in China.[16] As such, text messaging has the potential to be a scalable and powerful tool to deliver health information.[17] [18]

Prior studies of mobile phone text messaging have been conducted to improve glycaemic control,[19] hypertension,[20] medication adherence,[21] as well as to promote smoking cessation[22] and physical activity.[23] [24] These trials have contributed important knowledge, and some, such as The Tobacco, Exercise and Diet Messages (TEXT-ME) trial[25] and Trial to Examine Text-based mHealth for Emergency department patients with Diabetes (TExT-MED)[26] suggest that text messaging interventions can influence patient behaviours and improve risk profiles. Still, several questions remain about the generalisability of these findings, especially for populations from LMICs. Most trials to date have been designed to target a single condition; yet patients with cardiovascular disease usually manage multiple conditions, requiring several lifestyle and treatment recommendations. Additionally, most studies of text messaging interventions have not been grounded in behavioural change techniques (BCTs),[27] [28] which may influence the efficacy of the intervention. Finally, most prior studies have been limited to single-site interventions, many of which were underpowered and/or limited by selection bias. More data are needed to understand whether text messaging interventions should be adopted as an effective strategy for supporting cardiovascular disease prevention among diverse populations from LMICs.[29]

Accordingly, we designed and conducted the Cardiovascular Health And Text Messaging (CHAT) Study and CHAT-Diabetes Mellitus (CHAT-DM) Study. The primary objective of these two studies is to evaluate the efficacy of an automated text message-based intervention, based on BCTs, in improving risk factor control and adoption of healthy lifestyle behaviours among patients with known CHD, with or without DM, who were discharged from multiple hospitals throughout China.

## METHODS AND ANALYSIS
### Study overview

The CHAT and CHAT-DM studies are multicentre, single-blind, two-arm, randomised controlled trials of an automated mobile phone text message-based intervention with 6 months of follow-up. Patients were recruited from 37 hospitals across 17 provinces in China (figure 1, online supplementary material 1). The enrolment of participants began on 16 August 2016. The two studies were registered at http://www.clinicaltrials.gov (NCT02888769 and NCT02883842) accordingly. We retrospectively registered the trial 7 days later after enrolment of the first patient in CHAT Study , beyond the International Committee of Medical Journal Editors (ICMJE) recommendations of the journal, as we referred to the Food and Drug Administration (FDA) AA801 public law on the Clinicaltrials.gov website that clinical trials are registered no later than  21 days  after the first  patient  was enrolled. The recruitment was completed in April 2017, and the last follow-up visit is expected to finish in October 2017. All participants provided written informed consent at the initial trial visit.

### Study population

In CHAT, patients were eligible if they had established CHD defined as a history of AMI and/or percutaneous coronary intervention, access to a mobile phone to read and send text messages, and did not have diabetes. In CHAT-DM, patients were eligible if they had a history of documented CHD (as defined in CHAT) and diabetes, and had access to a mobile phone to read and send text messages. In both studies, patients were excluded if they could not read or send text messages, had cognitive or communication disorders, or could not provide informed consent. A 'screening log' of basic demographic information and reasons for not participating in patients deemed ineligible or declined to participate has been recorded.

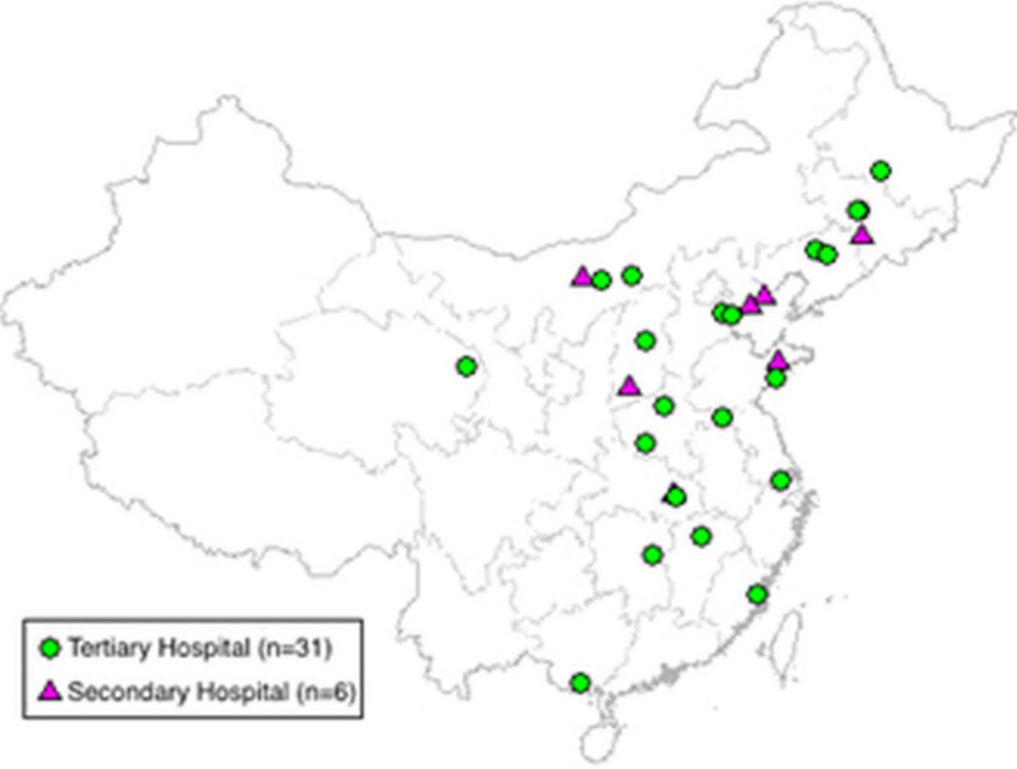

**Figure 1** Geographical distribution of sites in the Cardiovascular Health and Text Messaging (CHAT) Study and the CHAT-Diabetes Mellitus (DM) Study.

We recruited patients who had been hospitalised with CHD, with or without DM, and had medical records with definite discharge diagnoses available. The diagnoses of CHD and DM given by local physicians were adjudicated centrally, based on review of the patients' medical charts, which were sent to the China National Center for Cardiovascular Disease (NCCD).

### Randomisation and blinding

Participants were randomly allocated to either the intervention or the control arms in a 1:1 ratio using a computerised randomisation system. In order to achieve a balance of participants' characteristics in both arms, we employed a stratified randomisation approach, based on age, gender, AMI history, education degree and medical insurance type within each study. Researchers, statisticians and clinic staff were blinded to treatment allocation.

### Trial intervention

Participants in the intervention groups of the CHAT and CHAT-DM studies receive text messages about CHD risk factor modification for 6 months as well as standard treatment (described below). The control group in both studies receive two thank you text messages without risk factor modification support each month as well as standard treatment. A training session was held by research staff on enrolment to ensure that all participants were capable of receiving, reading and sending text messages on their mobile phones. Prior to commencement, a test message was sent to each participant to confirm the

phone number and that the system was working effectively. Whether the test messages had reached the patients would be recorded by local study staff, and if patients did not receive messages, local study staff would confirm and update the correct phone number. Participants were also informed that they could withdraw from the study by responding to a text message with a specific character. Researchers at NCCD could monitor the status of text message delivery, review responses sent by participants and manage withdrawal via a customised software platform. For instance, logs were kept to assess the proportion of text messages successfully delivered, the exact time of messages being sent and the reasons for those that failed to be sent.

### CHAT and CHAT-DM intervention development

A bank of 550 text messages (280 in CHAT and 270 in CHAT-DM) was developed by a multidisciplinary team of cardiologists, endocrinologists, psychologists, nurses and public health researchers using a three-phase systematic and iterative approach (figure 2). In the CHAT Study, text messages are categorised into five groups: (1) general education on CHD and AMI, (2) medication adherence, (3) blood pressure control, (4) physical activity and (5) smoking cessation. The text messages in the CHAT-DM Study cover a range of diabetes self-management topics including: (1) general education on CHD and DM, (2) medication adherence, (3) glucose monitoring and control, (4) blood pressure control, (5) physical activity

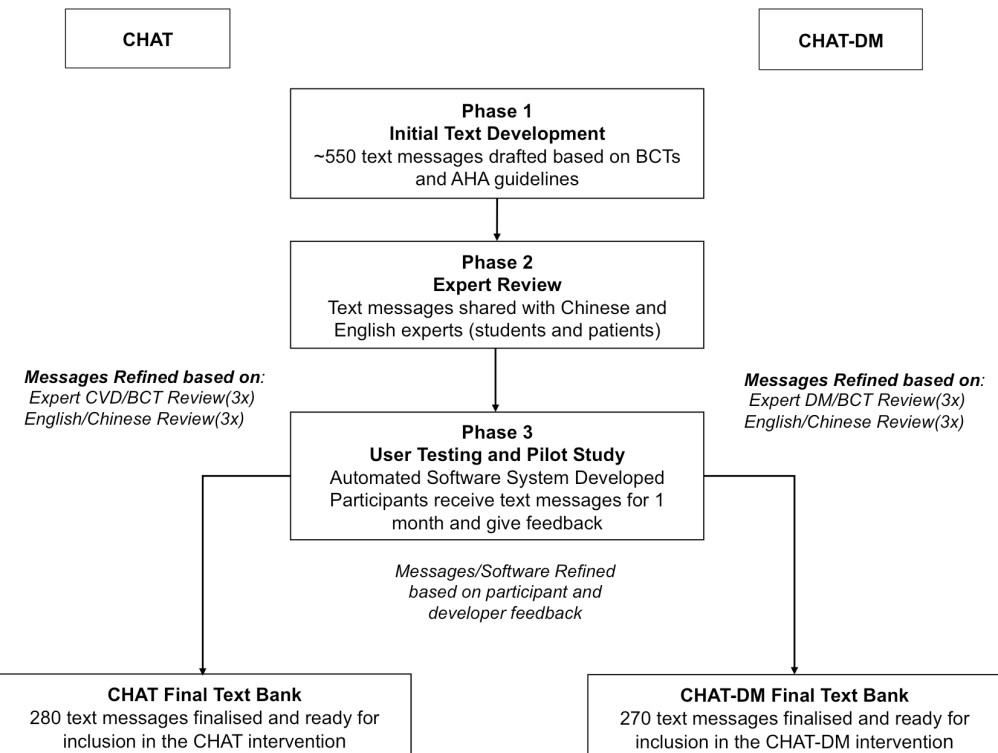

**Figure 2** Text development process. AHA, American Heart Association; BCT, behavioural change techniques; CHAT, Cardiovascular Health And Text Messaging; CVD, Cardiovascular Disease; DM, diabetes mellitus.

and (6) lifestyle recommendations such as diet and foot care. All text messages are in Chinese, and each text message is less than 70 Chinese characters, which would be equivalent to a 140–160-character message in English.

## Phase 1: Approach to developing text messages

Text messages were originally drafted by members of the research team in Chinese based on current guidelines[30–32] and standards of care pertaining to cardiovascular health and diabetes care. All messages were grounded in BCTs previously used to develop short health messages, providing information, goal setting, motivation, social support and stress management advice.[33] In order to strengthen designing and reporting of theory-based BCT interventions, we took into account the two recent BCT taxonomies by Michie and colleagues.[34 35] Efforts were devoted to selecting the BCTs most applicable to the Chinese cultural context and compatible with Chinese beliefs and values (table 1). For example, some text messages applied traditional Chinese aphorisms, chengyu and catchy rhyme schemes in order to make them more acceptable to patients. As another example, in designing text messages that were motivating, especially in challenging situations, the multidisciplinary group felt that Chinese people tend to prefer more direct and structured counselling instructions rather than indirect and insight-oriented approaches (a category of psychotherapeutic approaches that promote individuals' adaptive behaviour and course of action in life through understanding their internal motivation);[36] as such, text messages were written to provide practical approaches

and real life examples instead of abstract theories.[37 38] Social and family oriented goals were used more often than individual achievement to help improve health behaviours, consistent with cultural norms in China.[39 40] These messages were sent to experts in BCTs and counselling for review and were used as examples to draft the initial bank of 550 text messages. Once drafted, all messages were reviewed, critiqued and revised within the internal research team.

## Phase 2: Expert review

An expert panel made up of clinicians and academics reviewed each round of text message drafts with a different focus in each iteration. First, clinical experts (cardiologists, endocrinologists and psychologists) considered the accuracy, clarity and practical usefulness of each text message. Second, 21 messages from all categories in the CHAT and CHAT-DM Studies text banks were randomly selected and translated into English by bilingual researchers. Experts in the fields of cardiology, endocrinology, epidemiology, psychology and behavioural science in the USA reviewed them and provided suggestions for further refinement. Finally, Chinese researchers and a linguist reviewed all 550 text messages, further refined the language, and paid special attention to the cultural meaning of all text messages to ensure better understanding among the targeted population (elderly Chinese patients with cardiovascular disease, with and without diabetes). Once feedback from the experts was addressed, the text bank was updated and prepared for user testing.

**Table 1** Behaviour change techniques used in message development and example messages

| Behaviour change technique[33] | Content/explanation[33] | Example text message in English |
|---|---|---|
| Provide information about behaviour–health link | General information about behavioural risk | The way you cook can impact your health as well. Steaming, boiling and sautéing are better ways to cook than deep-frying and pan-frying. Cooking in less oil is a healthy alternative. |
| Prompt barrier identification | Identify barriers to perform the behaviour and plan ways of overcoming them | Taking diabetes medications and injecting insulin regularly can help control your blood sugar. Forgetting to take your medication? Try to set a repeating alarm on your cellphone to remind you to take your medication or insulin injection. |
| Set graded tasks | Set easy tasks, and increase difficulty until target behaviour is reached | Have you been finding it hard to quit smoking? In the beginning things are always hard. You can use a schedule to gradually reduce the number of cigarettes you smoke. For example, you may try going from 20 cigarettes to 15 per day for a week. |
| Provide instruction | Tell the person how to perform a behaviour and/or preparatory behaviours | If you experience symptoms of angina (severe chest pain), place one nitroglycerin tablet under your tongue. Sit, stay calm and rest if you ever forget medications while going out. If angina symptoms are not relieved within 10 min, seek medical attention immediately. |
| Prompt self-monitoring of behaviour | The person is asked to keep a record of specified behaviour(s) | A cold or diarrhoea will make your blood sugar levels rise, so monitor your glucose more frequently when you are sick. If you are using insulin, test your blood glucose six to eight times a day, keep a blood glucose log and share it with your health providers. |
| Prompt practice | Prompt the person to rehearse and repeat the behaviour or preparatory behaviours | As an old Chinese saying goes, 'It takes more than one cold day for a river to freeze three feet deep; ice in the river takes a long time to melt.' Similarly, cerebrovascular disease requires long-term prevention and treatment. Remember to take your medications as prescribed! |
| Plan social support or social change | Prompt consideration of how others could change their behaviour to offer the person help or (instrumental) social support | Quitting smoking on your own can be difficult. Tell your friends and family when you are quitting so that they will stop giving you cigarettes. Support and encouragement from your loved one can be helpful as well. |
| Stress management | May involve a variety of specific techniques (eg, progressive relaxation) that do not target the behaviour but seek to reduce anxiety and stress | Relaxation is something we need to learn and practice. Listening to music, reading or talking to friends and family can ease stress. |
| Motivational interviewing | Prompt the person to provide self-motivating statements and evaluations of their own behaviours to minimise resistance to change | Did you smoke less today than you did yesterday or days before? If you did reduce the amount of cigarettes, it is something worth celebrating. We are sure that you have put a lot of effort into quitting. Keep up the good work and you can make a difference! |

### Phase 3: User testing and pilot study

For user testing, 19 randomly selected text messages from the finalised text bank in the CHAT and CHAT-DM Studies were distributed to 39 individuals with CHD (with or without diabetes) for feedback. Likert-type items in a survey were used to assess usefulness and ease of understanding of the text messages. Open-ended questions were asked to obtain suggestions for improvement from participants. In total, 92.4% (582/630) rated the messages as easy to understand and 93.2% (587/630) rated the messages as useful. All 550 text messages were further modified based on feedback from user testing. Results of the scores for the message categories in this phase are summarised in table 2.

Next, the refined text message banks underwent pilot testing to evaluate the efficacy of the message delivery system and quality of the user experience. A customised software programme sent messages through a gateway interface, allowing them to be sent to all individuals'

mobile phones free of charge and at a bulk-rate cost (¥0.1 per message, or US$0.01 USD) to the research team. Participants from the user testing were enrolled in the pilot testing after providing written informed consent (n=33). For the pilot study, messages drawn from the entire text bank were sent to participants. At the end of the 1-month pilot study, 30 participants reported their experience of receiving messages and commented on frequency, timing, content and potential impact of text messages. Minor changes were made to the text message bank following the pilot study based on this feedback.

### Frequency and timing of text message delivery

Each participant in the intervention group of the CHAT and CHAT-DM Studies receives six text messages per week, randomly selected by the software system, across a 6-month timeframe. The messages are sent at one of three random times (9:00, 12:00 or 4:00), during weekends and weekdays (excluding Monday to give people

**Table 2**  Survey scores for various message categories in user test

| | Strongly agree or agree (%) | Neutral (%) | Strongly disagree or disagree (%) |
|---|---|---|---|
| General health messages | | | |
| Information easy to understand (n=142) | 130 (91.6) | 6 (4.2) | 6 (4.2) |
| Information was useful (n=142) | 129 (90.9) | 9 (6.3) | 4 (2.8) |
| Hypertension | | | |
| Information easy to understand (n=181) | 168 (92.8) | 7 (3.9) | 6 (3.3) |
| Information was useful (n=181) | 169 (93.4) | 6 (3.3) | 6 (3.3) |
| Medication adherence | | | |
| Information easy to understand (n=149) | 134 (89.9) | 10 (6.7) | 5 (3.4) |
| Information was useful (n=149) | 137 (92.0) | 10 (6.7) | 2 (1.3) |
| Physical activity | | | |
| Information easy to understand (n=116) | 109 (93.9) | 6 (5.2) | 1 (0.9) |
| Information was useful (n=116) | 111 (95.7) | 4 (3.4) | 1 (0.9) |
| Smoking cessation | | | |
| Information easy to understand (n=42) | 41 (97.6) | 1 (2.4) | 0 (0) |
| Information was useful (n=42) | 41 (97.6) | 0 (0) | 1 (2.4) |

a break). The text messages were semipersonalised with participants' preferred name at the beginning of some messages, and the smoking status being the second factor considered for personalisation. In the CHAT Study, participants receive two general education messages on CHD (one if an active smoker), two blood pressure control messages, one medication adherence message and one physical activity message per week. Smokers receive one smoking cessation message per week. Participants in the CHAT-DM intervention group receive one general message, one blood pressure related message, one glucose control message, one lifestyle modification message, one medication adherence message and one physical activity message per week (table 3).

Initially, a test message was sent to each participant to ensure that the correct mobile phone number had been recorded and the system was functioning appropriately. All participants received a personalised welcome message, a follow-up reminder and a birthday greeting while enrolled in the study. Most of the text messages were developed to be unidirectional and participants are not anticipated to reply, however bidirectional text messages, checking on medication adherence and blood pressure/glucose level measurements are sent at weekly intervals to evaluate patient engagement. Throughout the 6-month follow-up, research staff call participants if they do not respond for two consecutive weeks, and only one call is made per patient during the intervention period, so as not to confound the intervention.

## Procedures for data collection and management
### Data collection
Basic and contact information (ID number, address and phone number), anthropometric data (waist circumference, height and weight), resting blood pressure, heart rate, ambulatory blood pressure, and information on socioeconomic status, risk factor control and current medications were collected at baseline at hospital recruitment sites. Detailed information on patient outcomes (hospitalisations, discharge diagnoses, etc) was also collected during the survey and hospital records or death certificates obtained where necessary for adjudication. Additional assessments of baseline medication adherence, physical activity (International Physical Activity Questionnaire: IPAQ),[41 42] Cardiovascular disease (CVD)-specific health status (Seattle Angina Questionnaire: SAQ),[43] health status (EuroQol five-dimensional questionnaire: EQ-5D)[44] and smoking status were also conducted in person. Lastly, blood and urine samples were collected for local lab tests and eventual transfer to the core lab in Beijing at the biobank of NCCD. To ensure the standardisation and accuracy of the sample analysis results, we developed standard operating procedures and trained local study researchers repeatedly regarding samples collection, separation, storage and transfer process. Blood tests, including low-density lipoprotein cholesterol (LDL-C), glycaemic haemoglobin ($HbA_{1C}$) and fasting blood glucose (FBG) will be analysed at the central laboratory. Research staff would collect the above information again as listed in table 4 at the follow-up visit. We conducted on-site monitoring of recruitment, physical measurements, sample collection and document completeness (eg, informed consent) by trained staff from NCCD to ensure the quality of data collection.

### Data management
A proprietary software platform was developed by the study Information Technology (IT) team for use in sending text messages to participants. The platform is capable of sending tailored and semipersonalised text

**Table 3** Example of text messages for the CHAT and CHAT-DM Studies

| CHAT text (six texts/week) | CHAT-DM text (six texts/week) |
|---|---|
| **General education (CVD) (2x/week)*** <br> The most common risk factors for coronary artery disease are smoking, obesity, high blood pressure, high cholesterol and diabetes. However, most of them can be controlled in an appropriate way. | **General education (DM) (1x/week)** <br> Diabetes is not terrible and there are many things you can do to prevent problems from diabetes, such as monitoring blood glucose, watching your diet, keeping fit and taking pills regularly. |
| **Blood pressure control (2x/week)** <br> Most people do not experience any symptoms of high blood pressure. Do not stop taking blood pressure medication unless directed by your doctor. It is important for patients with hypertension to take medication diligently and to monitor their blood pressure on a regular basis. | **Blood pressure control (1x/week)** <br> Home blood pressure monitoring is highly recommended! You can get an accurate picture of your heart health and understand daily changes in blood pressure, which is helpful for doctors to adjust medications for you. |
| **Medication adherence (1x/week)** <br> Do you have a problem remembering to take your blood pressure medications? If so, try to tell your family about your medicine schedule so they can remind you. | **Medication adherence (1x/week)** <br> Talk to doctors about your concerns and any uncomfortable symptoms after taking pills. Let your doctor help you to find the right medication for you. |
| **Physical activity (1x/week)** <br> You can still choose low-intensity exercise even after heart attack, such as walking and t'ai chi, at a slower pace and stick to your exercise plan. Always consult your physician before beginning any exercise programme. | **Physical activity (1x/week)** <br> Try brisk walking—a convenient, safe and cost-effective way of exercising! It's good for your heart and will help control blood glucose. |
| **Smoking cessation (1x/week)†** <br> Do you worry about your family having health problems because of your smoking? Quitting is an important choice you can make to benefit your family's health, too. Secondhand smoke can cause respiratory disease, lung cancer and heart disease. | **Diabetes management (1x/week)** <br> See a doctor before you travel. Always carry your diabetes medications and insulin, glucose meter and strips, so that you can better monitor your glucose. Carry some hard candy and crackers to avoid low sugar. |
| | **Lifestyle intervention (1x/week)** <br> Individuals with diabetes should consume a balanced diet, and eat smaller but more frequent meals. Consider splitting your meal and save it for a snack later. Some healthy snack choices include tomatoes, cucumbers and sugar-free biscuits. |

*In CHAT, non-smokers receive two general education messages per week, while smokers only receive one per week.
†In CHAT, only smokers receive smoking cessation messages.
CHAT, Cardiovascular Health And Text Messaging; CVD, cardiovascular disease; DM, diabetes mellitus.

**Table 4** Baseline and follow-up data collection

| Information | Baseline | 6 months |
|---|---|---|
| Basic and contact information | √ | √ |
| Physical examination: BP, HR, waist circumference, weight, height | √ | √ |
| Ambulatory blood pressure | √ | √ |
| Outcome (death, myocardial infarction, angina, stroke, revascularisation, etc) | √ | √ |
| Hospitalisations | √ | √ |
| Current medications | √ | √ |
| Medication adherence | √ | √ |
| Physical activity (International Physical Activity Questionnaire) | √ | √ |
| CVD functional status (Seattle Angina Questionnaire) | √ | √ |
| Health status (EuroQol five-dimensional questionnaire) | √ | √ |
| Socioeconomic status | √ | √ |
| Risk factors control | √ | √ |
| Urine cotinine/nicotine test | √ | √ |
| Blood and urine samples for core lab and local test | √ | √ |

BP, blood pressure; CVD, cardiovascular disease; HR, heart rate.

messages to participants and also recording responses. Additionally, this web-based platform is used to monitor project progress, as well as provide management support for hospitals, staff members, equipment and sampling collection and transfer. Trained medical staff members fill out predesigned onscreen case report forms at each site, and data are then securely transmitted to the central server through automatic electronic transfer. To ensure the reliability and validity of the data, continuous checks are run to ensure that data being entered are complete and meet predefined data formats and ranges. The database is regularly backed up and password protected so that only a limited number of approved staff members can access the data. In order to ensure the confidentiality of all personal information, data confidentiality policies of NCCD on data collection, storage and analysis have been strictly imposed.

## Outcomes

In the CHAT Study, the primary outcome is the change in systolic blood pressure (SBP) after 6 months. Secondary outcomes include a change in proportion of patients achieving a SBP <140 mm Hg, change in proportion of non-smokers, change in medication adherence , as well as change in plasma mean level of LDL-C, change in level of body mass index (BMI) and change in level of physical activity. Exploratory outcomes include the prognosis of the patients at 6 months, such as death, non-fatal myocardial infarction, stroke and any rehospitalisation, as well as health status measured by SAQ and EQ-5D.

In the CHAT-DM Study, the primary outcome is the change in glycaemic $HbA_{1C}$ as measured by the central blood sample. Secondary outcomes include a change in proportion of patients achieving $HbA_{1C}$<7%, change in medication adherence, as well as change in mean level of FBG, SBP, LDL-C, BMI and physical activity. Exploratory outcomes include prognosis of patients at 6 months, including death, non-fatal myocardial infarction, stroke and any rehospitalisation, as well as health status (SAQ and EQ-5D).

Blood pressure is measured on the right upper arm after 5 min of rest in a seated position using an electronic blood pressure monitor (Omron HEM-7111; Omron Corporation, Dalian, China). Two measurements are taken and the mean value is calculated. If the difference between the two SBP or diastolic blood pressure readings is larger than 5 mm Hg, a third measurement is done, and the mean value of the three readings is calculated.[45 46] $HbA_{1C}$ is determined using a high-performance liquid chromatography technique with ADAMS $A_{1C}$HA-8180 (ARKRAY, Japan). BMI is calculated by dividing weight in kilograms by height in metres squared. Physical activity is measured in metabolic equivalents of task per minute per week, using the short version of IPAQ.[41 42] Smoking status is determined using either self-reported smoking status or urine-cotinine testing strip with a cut-off of 200 ng/mL cotinine (COT Cotinine Test Colloidal Gold; Hangzhou Clongene Biotech, China).[47 48] Quality of life is measured using the short version of SAQ[43] and EQ-5D.[44] Local study staff obtained information of baseline hospitalisation, readmissions to hospitals and death during the patient's interview, with medical records, death certificates or death records collected as supporting documents. If the patient died at home without any evidentiary material, a structured summary of death conversation with family members would be reported. All information was sent to NCCD for central adjudication according to prespecified criteria by trained clinicians. If the patient was rehospitalised in other hospitals, for example, the study investigators will contact the specific hospital, copy those medical records and transmit them to NCCD as required.

## Statistical analysis

Intervention evaluation will be carried out on an intention-to-treat basis. Values of analysed parameters at baseline between intervention group and control group will be compared using Student's t-tests for continuous variables or $\chi^2$ tests for categorical variables according to the analysis plan. Mann-Whitney U tests will be used where continuous data are not normally distributed. The primary analysis will employ analysis of covariance with baseline values of the analysed end points used as covariates when appropriate. The mean change of each risk factor will also be compared between groups. In all analyses, 95% CIs and two-sided P values will be reported. We will conduct a prespecified second analysis with

adjustment for patient characteristics as well as subgroup analyses based on age, sex, education, smoking status and tertile levels of end points.

All sample size calculations in both studies are for 80% power and 0.05 (one-sided) level of significance, allowing for 20% dropout rate during follow-up. In the CHAT Study, we estimate a mean SBP level of 132 mm Hg (SD, 18 mm Hg) in the study population according to the report of previous studies,[49] and assume an absolute difference in SBP changes of 5 mm Hg at 6 months from baseline in the intervention group and no change in the control group. A total sample size of 800 patients would be adequate to detect this difference in SBP changes between two groups even when considering the potential non-compliance to the intervention. In the CHAT-DM Study, we assume a mean $HbA_{1C}$ level of 7.2% (SD, 1.6%) based on data from studies involving similar populations[50 51] and supposed a 0.5% absolute difference in $HbA_{1C}$ changes in treatment group and no change in the control group at 6 months from the baseline. A total sample size of 500 patients is adequate to detect this difference in $HbA_{1C}$ change between two groups even when considering the potential non-compliance to the intervention. The trial results will be reported in accordance with the Standard Protocol Items: Recommendations for Interventional Trials (SPIRIT) checklists.

## DISCUSSION

The CHAT and CHAT-DM Studies aim to assess the efficacy of an innovative intervention for improving secondary prevention of CHD by using simple and cost-effective text messaging technology among patients with known CHD, with or without DM, throughout China. To the best of our knowledge, these two trials are the first to investigate the efficacy of text messages to support management of CHD and DM among a large population in China, and may set a model of such interventions that can be leveraged to improve risk factor management in resource-constrained settings.

The CHAT and CHAT-DM Studies have several strengths. While prior studies evaluated the effectiveness of mobile phone text messaging to improve single individual health behaviours, very few trials have addressed the role of text messages in managing multiple risk factors in patients with CHD or DM, which is a reality for many patients. Patients with CHD and DM often have multiple risk factors, yet the care for these conditions is often fragmented. Targeting multiple risk factors concurrently instead of managing a single risk factor may be more efficient and impactful on disease management as this strategy is more patient-centred than disease-centred, and have a greater likelihood of improving risk factor control and cardiovascular outcomes.[52 53] Additionally, there is currently little evidence about the effectiveness of such interventions for high-risk patients with CHD and diabetes.

The CHAT and CHAT-DM Studies are further distinguished by their large sample sizes and culturally appropriate, theory-driven text messages. Most text message intervention studies that have focused on health behaviour or diabetes management have had sample sizes ranging from 18 to 357 participants.[54] CHAT has a sample size of around 800 while the sample size of CHAT-DM is around 500. It is noteworthy that CHAT is the first study using theory-based text messaging as a method of delivery for behaviour change of patients with CHD in China. Data from other text messaging intervention studies suggest that text messaging programmes achieved better results when message content is theory-based,[54] however, few text messaging studies specified a theoretical rationale. Further, text messages were tailored to a specific Chinese patient population. Studies have found that the majority of Chinese adults prefer learning by following directive rules and guidelines,[55] and practical counselling instructions were provided through text messages to help them better modify health behaviours. Apart from this, the text message language was designed to be plain and easy to memorise, consistent with Chinese cultural features such as aphorism, chengyu and catchy rhyme schemes, making it more acceptable to patients.

The two studies have several additional strengths. In contrast to most prior single-centre text messaging trials, these two studies are conducted at 37 participating sites spread across a large and geographically diverse country. These results may be more generalisable. Moreover, our research team devoted significant attention to quality data collection, including rigorous on-site monitoring of questionnaire answers, samples collection and data management. Furthermore, an evaluation of the acceptability and feasibility of our text message-based intervention will provide further important evidence to inform future studies, particularly with regards to optimising text content, text frequency and the overall user experience.

Our study also has some potential limitations. First, medication adherence and physical activity are measured by self-report, which carries the possibility of recall bias and social desirability bias. However, we believe that any such bias would be balanced across the treatment and control groups, and IPAQ have been validated and are widely used in measuring these metrics. Second, the text messages, though semipersonalised (participant's preferred name, smoking status) were not tailored specifically to every individual.

The CHAT and CHAT-DM Studies have important public health implications. As low cost, non-pharmacological interventions, the CHAT and CHAT-DM Studies may serve as important models for patient-centred, evidence-based public health interventions. LMICs, including China, face challenges with huge burdens of CVD and DM, limited health resources, and geographically and culturally diverse patient populations, making them the ideal places to conduct such studies. Still, high levels of mobile phone ownership across countries and income levels signal a promising new avenue for clinical research

and that it may be possible to scale effective mobile health interventions for delivery to large populations in the coming years. Text messaging interventions are simple, cost-effective and incur minimal, if any, risk. Hence, if our interventions are effective in improving risk factor control and lifestyle modification, large-scale implementation could benefit a diverse population in China.

In conclusion, the CHAT and CHAT-DM Studies are multicentre, randomised controlled trials that are being conducted at 37 hospitals, and will thoroughly test the efficacy of text messages to support secondary prevention for CHD and DM in China. These two trials target high-risk patient populations with a culturally sensitive, scalable and cost-effective intervention, and have the potential to provide novel insights into disease management and can be scaled up to improve health in a significant proportion of patients in the future.

**Author affiliations**
[1]China Oxford Centre for International Health Research, Fuwai Hospital, National Center for Cardiovascular Diseases, Chinese Academy of Medical Sciences and Peking Union Medical College, Beijing, China
[2]Center for Outcomes Research and Evaluation, Yale University/Yale-New Haven Hospital, New Heaven, Connecticut, USA
[3]Health Outcomes Research, Saint Luke's Mid America Heart Institute/University of Missouri-Kansas City, Kansas, Missouri, USA
[4]Division of Cardiology, University of Colorado Anschutz Medical Campus, Aurora, Colorado, USA

**Acknowledgements** The authors thank the study teams at the China Oxford Centre for International Health Research and the Yale-New Haven Hospital Center for Outcomes Research and Evaluation for the multiple contributions in the areas of study design and operations, particularly the contributions to the text message bank by Xuekun Wu, Si Xuan and Xiuling Wang, and data collection and analysis by Ying Sun, Chaoqun Wu, Xueke Bai, Jiamin Liu and Wuhanbilige Hundei. The authors also thank Clara Chow and her team in Sydney, Weigang Zhao, Geng Liu, Yuanlin Guo, Zhuo Xu and Zengwu Wang for their valuable advice. The authors also thank the Chinese government for the support provided.

**Contributors** XH, QD, PH, XZ, MLI, WG, JL, XL, LJ: study concept and design; XH, ESS, QD, PH, CM, XZ, HZ, XY, JAS, FAM, HMK: drafting the initial manuscript and critical revision of the paper. All authors read and approved the final manuscript.

**Funding** This project was partly supported by the Research Special Fund for Public Welfare Industry of Health (201202025, 201502009) from the National Health and Family Planning Commission of China, the National Key Technology R&D Program (2015BAI12B01, 2015BAI12B02) from the Ministry of Science and Technology of China, CAMS Innovation Fund for Medical Sciences (CIFMS 2016-I2M-2-004), and the 111 Project (B16005). ESS is supported by grant K12HS023000 from the Agency for Healthcare Research and Quality Patient-Centered Outcomes Research Institute (PCORI) mentored career development programme. The sponsors had no role in the preparation or approval of the manuscript.

**Competing interests** HMK is a recipient of research agreements from Medtronic and Johnson & Johnson (Janssen) through Yale University to develop methods of clinical trial data sharing. He is also the recipient of a grant from the Food and Drug Administration and Medtronic to develop methods for postmarket surveillance of medical devices; and is the founder of Hugo, a personal health information platform. All other authors have no conflicts of interests to declare.

**Patient consent** Obtained.

**Ethics approval** The central ethics committee at the China National Center for Cardiovascular Disease (NCCD) and the Yale University Institutional Review Board approved the CHAT and CHAT-DM studies. All collaborating hospitals accepted the central ethics approval except for 8, which obtained local approval by internal ethics committees.

**Provenance and peer review** Not commissioned; externally peer reviewed.

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
