## [Reviewer comments · BMJ Open]

ARTICLE DETAILS

TITLE (PROVISIONAL)	Design and rationale of the Cardiovascular Health and Text Messaging (CHAT) Study and CHAT-Diabetes Mellitus (CHAT-DM) Study: two randomized controlled trials of text messaging to improve secondary prevention for coronary heart disease and diabetes
AUTHORS	Huo, Xiqian; Spatz, Erica; Ding, Qinglan; Horak, Paul; Zheng, Xin; Masters, Claire; Zhang, Haibo; Irwin, Melinda; Yan, Xiaofang; Guan, Wenchi; Li, Jing; Li, Xi; Spertus, John; Masoudi, Frederick; Krumholz, Harlan; Jiang, Lixin

VERSION 1 – REVIEW

REVIEWER	Lijing Yan Duke Kunshan University, China
REVIEW RETURNED	28-Jul-2017

GENERAL COMMENTS	This protocol paper describes two related studies: CHAT and CHAT-DM in China. The interentions have the potential to be effective, scalable, and low-cost for secondary prevention of CVD and DM. The studies are well designed and the paper is very well written. The design of the study is thoughtful, valid, and represents improvements over published studies such as theory-based and culturally tailored. Limitations of the studies are appropriately acknowledged and can be addressed in future studies such as individualization and interactivity of messages. Recommend accept with minor revisions suggested as follows: • The study began in August 2016, nearly 1 year ago and the duration of intervention is 6 months. Even considering staged roll-out across sites, the baseline or even most of the intervention may be completed. Consider presenting baseline data in this paper or at least disclose the timeline/progress status of the study.• Though involving many centers is a major strength, quality control to ensure standardization of protocol implementation including patient selection, enrollment, and data collection becomes critically important. This paper should include a description on quality assurance across sites. The intervention is centrally delivered (page 14 line 31), thus alleviates such concern.• Page 14 lines 24-26 and lines 40-44: Data collection of some outcomes (though exploratory) need to be clarified in a concise way, namely patient prognosis at follow-up (death, MI, stroke, re-hospitalization).
---

	Is it simply through hospital records? How about prognosis not captured in the study hospitals? Or through self-report? How about death?  • Page 15, line 17: Description of secondary outcomes lacks precision. We expect that all outcomes (LDL, BMI, etc) be defined in a way that can generate "proportion" (like SBP<140 mmHg) since the proportions are the outcomes. • Page 16: line 49: Define the pre-specified subgroups. • Page 37, line 17: Table 4: "Outcome" is listed as a separate measure, which is too vague.
--	--

REVIEWER	Andre Matthias Müller Saw Swee Hock School of Public Health, National University of Singapore, Singapore
REVIEW RETURNED	10-Aug-2017

GENERAL COMMENTS	Thank you for the opportunity to review this study protocol. I enjoyed reading the manuscript which describes a thoroughly developed mobile health intervention. You have considered the cultural context in which the intervention is rolled out and conducted user testing. It is great to see that mobile health initiatives are becoming part of the research landscape in China-I am looking forward to see the results. I made a number of comments mainly pertaining to language, flow and clarity of the text. Please see my comments below: Overall: I wonder if you expect a bit too much from the patients?! You target a number of health behaviours and also expect patients to monitor certain things (blood pressure etc). Most people that will enrol in the study will likely have a not so health lifestyle (prior to the study) and there is some research indicating that such people could be overwhelmed by the things they are asked to do. As a result, they might give up early on. Have you considered this? Overall: I wonder what you mean by semi-personalised text messages? Strengths and Limitations (first): I suggest you say "intervention" instead of "solution". Strengths and Limitations (fourth): I suggest you split up the sentence-after "country". Introduction: I am not sure what you mean by this sentence: "Prior studies reveal that only one quarter of hospitalized patients do not fill all...". Do you mean that only one quarter obtain medications from their prescriptions? Please make this sentence easier to digest. Introduction: Page 5, line 38: Could you use another word instead of "influential". Maybe say "...are a barrier".
---

Introduction: Page 5, line-54-55: Could you rewrite this: "Mobile phones already serve as a primary, inexpensive and quick form of communication, alerts and reminders." You could say "Mobile phones are already a primary and inexpensive form of communication, and they are also used to schedule alerts and reminders".

Introduction: Page 4-5, line-57-7: Could you break up this sentence? I suggest to add the text message information (200,000 messages) to the end of the paragraph as you talk about text messages there.

Introduction: Page 5, line 50-51: Could you restructure the sentence? I think what you mean is that mobile phones, that are ubiquitous in people's lives can be used to deliver interventions that help people to adopt secondary prevention strategies.

Introduction: page 6, when talking about LMIC studies: Consider adding these 2 very relevant references: Müller, A. M., Alley, S., Schoeppe, S., & Vandelanotte, C. (2016). The effectiveness of e- & mHealth interventions to promote physical activity and healthy diets in developing countries: A systematic review. *International Journal of Behavioral Nutrition and Physical Activity*, 13(1), 109.; Hall, C. S., Fottrell, E., Wilkinson, S., & Byass, P. (2014). Assessing the impact of mHealth interventions in low-and middle-income countries—what has been shown to work? *Global health action*, 7(1), 25606.

Methods (under Randomization and Blinding): page 8: Delete the extra full stop after last sentence.

Methods (under trial intervention): page 8, line 51-54: I wonder why you say that the control group received Thank You messages without medical information. I thought the intervention group receives advice on risk factor modification—this doesn't need to mean it is medical (e.g., advice on physical activity is not really medical). So, I suggest you say that the CG did not receive risk factor modification support.

Methods (under trial intervention): page 9, line 7-10: Could you restructure this sentence "Participants were also instructed how to withdraw from the study should they desire, by responding with a specific character" to make it easier to digest? You could say "Participants were also informed that they could withdraw from the study by responding to a text message with a specific character."

Methods (CHAT and CHAT DM intervention development): page 9, last sentence: Could you change to "... to a 140-160 character message in English"?

Methods (under Phase 1 Approach to...): I just want to let you know that there are two more updated behaviour change technique taxonomies: Michie, S., Ashford, S., Sniehotta, F. F., Dombrowski, S. U., Bishop, A., & French, D. P. (2011). A refined taxonomy of behaviour change techniques to help people change their physical activity and healthy eating behaviours: the CALO-RE taxonomy. *Psychology & health*, 26(11), 1479-1498.; Michie, S., Richardson, M., Johnston, M., Abraham, C., Francis, J., Hardeman, W., ... & Wood, C. E. (2013). The behavior change technique taxonomy (v1) of 93 hierarchically clustered techniques: building an international consensus for the reporting of behavior change interventions. *Annals of behavioral medicine*, 46(1), 81-95. This is purely for your information in case you are interested.

Methods (under Phase 1 Approach to...): I am very happy that you did this: "Efforts were devoted to selecting the behavioral change techniques most applicable to the Chinese cultural context and compatible with Chinese beliefs and values (Table 1)." You explain what you did and why you did it (considering culture and norms). This is helpful for my own work.

Methods (Table 1): I found some typos in the table-please correct.

Methods (Table 1): I generally like the text messages. Just two comments. 1. Sometimes you could give a stronger rationale why a behavior might be good (e.g., message 1 in table). Saying that something is good for health is a bit unspecific and many people might want to know more to also look out for changes (e.g., for physical activity you could say that it might get easier to climb stairs). 2. You sometimes use complicated words in the text messages and I wonder if people will be confused by this? We know that the reading age of the general population is not very high (about 11 years in the UK). However, as you piloted the messages I believe you have checked that. It might also be a language issue here. The messages might be easier to understand in Chinese (for Chinese speakers).

Methods (under Phase 2 Expert review): Page 11, line 8-13: I wonder what you mean by "clinical benefit of each text message"? I don't see that a message can have a clinical effect, especially when I look at your outcomes. Can you clarify this in the text?

Methods (under Phase 3 User testing and pilot study): I don't understand how the numbers for the user testing add up. You say you checked 19 text messages with 39 people ($19 * 39=741$). However, you also indicate that 630 was the total. Would be great if you could say what was actually calculated and how.

Methods (under data collection): Page 14, line 21-25: Please rewrite this sentence to make it easier to understand.

Methods (Table 4): What does the variable "Outcome" mean (there are ticks for it).

Methods (under data management): Page 14, line 38-43: Please rewrite this sentence to make it easier to understand (esp., last part).

Methods (under outcomes): Page 15, line 14: I suggest to add a "the" after "In" (In the CHAT study). Same for the CHAT DM study.

Methods (under statistical analysis): I wonder why you measure a number of variables but do not plan to control for them (e.g., age). Also, I suggest to use an ANCOVA on the change scores (baseline as one covariate).

Methods (under statistical analysis): Page 16, line 49-51: Please reword this sentence as it seems to not be correct.

Methods (under statistical analysis): Why do you think the SBP reduction in the intervention group will be smaller?? Isn't the goal of the intervention to reduce this more than in the control group?

Methods (under statistical analysis): I am not sure I understand your sample size calculation correctly. I assume you want to have a difference in change between the 2 groups of 3.5mm HG? Is that correct?

Discussion: Page 18, line 10: I suggest you use "intervention" instead of "solution".

Discussion: Page 18, line 32-41: You make an argument for interventions that target multiple risk factors. As I indicated earlier, I am not sure that targeting multiple risk factors at once is a good idea as people might be overwhelmed. In your argument, I don't understand why you think this approach is patient-centred (please correct from "centric" to "centred")?? The references you cite are fairly old-in this study promoting many health behaviours did not work: Reinwand, D. A., Schulz, D. N., Crutzen, R., Kremers, S. P., & de Vries, H. (2015). Who follows eHealth interventions as recommended? a study of participants' personal characteristics from the experimental arm of a randomized controlled trial. *Journal of medical Internet research*, 17(5). You could rephrase your Discussion slightly to give a more balanced view.

Discussion: Page 19, line 32-39: I suggest you split up this sentence. The part "...the results" could be a new sentence to make your point more prominent.

Discussion: Page 20, line 5-7: After "validated" add "are" (widely used), and say "these metrics".

Discussion: Page 20, line 8-12: You say here that the messages were not tailored, but earlier you said they were. Please be consistent. For me, it seems as if the messages were generic/not tailored.

Discussion: Page 20, line 37-44: Consider rewording this sentence to make it easier to understand.

Discussion: Page 20, line 53: You say the RCT will check the efficacy of text messages in LMICs. Please reconsider this statement: You only run the study in China. You cannot generalise to other LMICs (especially, those that are very different compared to China).

Discussion: last sentence: 1-You use past tense here (targeted), but the study is still under way. So, you could say "targets". 2-break up this sentence to make it easier to understand. I suggest a full stop after "intervention". 3-may just say that the intervention can potentially benefit a significant portion of the population (instead of "over thousand").

VERSION 1 – AUTHOR RESPONSE

Reviewer #1

This protocol paper describes two related studies: CHAT and CHAT-DM in China. The interventions have the potential to be effective, scalable, and low-cost for secondary prevention of CVD and DM. The studies are well designed and the paper is very well written. The design of the study is thoughtful, valid, and represents improvements over published studies such as theory-based and culturally tailored. Limitations of the studies are appropriately acknowledged and can be addressed in future studies such as individualization and interactivity of messages.

Recommend accept with minor revisions suggested as follows:

Comment 1: The study began in August 2016, nearly 1 year ago and the duration of intervention is 6 months. Even considering staged roll-out across sites, the baseline or even most of the intervention may be completed. Consider presenting baseline data in this paper or at least disclose the timeline/progress status of the study.

Response: Considering the limited word count and space, we plan to present baseline data in the main paper of study. As suggested by the reviewer, we have included a timeline of the status of the study as per below:

Methods and Analysis (page 7, paragraph 2, lines 22-23): “The recruitment was completed in April 2017, and the last follow-up visit is expected to finish in October 2017.”

Comment 2: Though involving many centers is a major strength, quality control to ensure standardization of protocol implementation including patient selection, enrollment, and data collection becomes critically important. This paper should include a description on quality assurance across sites. The intervention is centrally delivered (page 14 line 31), thus alleviates such concern.

Response: We have provided a more detailed description on quality assurance. Regarding patient selection, we added a detail on diagnosis confirmation as follows:

Methods and Analysis (page 8, paragraph 1, lines 13-17): “We recruited patients who had been hospitalized with CHD, with or without DM, and had medical records with definite discharge diagnoses available. The diagnoses of CHD and DM given by local physicians were adjudicated centrally, based upon review of the patients’ medical charts, which were sent to the China National Center for Cardiovascular Disease (NCCD).”

Regarding enrollment, we have employed several approaches to ensure study quality. First, to make sure that all participants being enrolled had access to a mobile phone to read and send text messages, study staff confirmed patients’ capability before enrollment. Moreover, as mentioned in the paper on page 9, line 9, a training session was held by research staff upon enrollment to ensure that all participants could receive, read and send text messages on their mobile phones. Second, to confirm the correct phone number and that the system was working effectively, a test message was sent. Our detailed description of these processes in the paper is as follows:

Methods and Analysis (page 9, paragraph 2, lines 11-16): “Prior to commencement, a test message was sent to each participant to confirm the phone number and that the system was working effectively. Whether the test messages had reached the patients would be recorded by local study staff, and if patients did not receive messages, local study staff would confirm and update the correct phone number.”

Regarding data collection, we modified the descriptions about sample management, and also added two details on page 14 and 15 as follows:

Methods and Analysis (page 15, paragraph 1, lines 1-6): “To ensure the standardization and accuracy of the sample analysis results, we developed standard operating procedures and trained local study researchers repeatedly regarding samples collection, separation, storage and transfer process. Blood tests, including low-density lipoprotein cholesterol (LDL-C), glycemic hemoglobin (HbA1C) and fasting blood glucose (FBG), will be analyzed at central laboratory.”

Methods and Analysis (page 15, paragraph 1, lines 7-10): “We conducted on-site monitoring of recruitment, physical measurements, sample collection and document completeness (e.g. informed consent) by trained staff from NCCD to ensure the quality of data collection.”

Additionally, as mentioned in the Data Management section (page 15, paragraph 2, lines 18-23), trained medical staff members administered the interviews using an electronic data capture platform, in which continuous checks are run to ensure that data are complete and meet predefined formats and ranges. Moreover, as kindly pointed out by the reviewers, the intervention is centrally delivered and thus can help ensure standardization of protocol implementation.

Comment 3: Page 14 lines 24-26 and lines 40-44: Data collection of some outcomes (though exploratory) need to be clarified in a concise way, namely patient prognosis at follow-up (death, MI, stroke, re-hospitalization). Is it simply through hospital records? How about prognosis not captured in the study hospitals? Or through self- report? How about death?

Response: We thank the reviewer for the comments, and we have added detailed descriptions of outcome collection as per below:

Methods and Analysis (page 17, paragraph 1, lines 14-23): “Local study staff obtained information of baseline hospitalization, readmissions to hospitals and death during the patient’s interview, with medical records, death certificates or death records collected as supporting documents. If the patient died at home without any evidentiary material, a structured summary of death conversation with family members would be reported. All information was sent to the NCCD for central adjudication according to pre-specified criteria by trained clinicians. If the patient was re-hospitalized in other hospitals, for example, the study investigators will contact the specific hospital, copy those medical records and transmit them to NCCD as required.”

Comment 4: Page 15, line 17: Description of secondary outcomes lacks precision. We expect that all outcomes (LDL, BMI, etc) be defined in a way that can generate “proportion” (like SBP<140 mmHg) since the proportions are the outcomes.

Response: In the CHAT study, we assess changes in the proportion of patients that achieve a SBP<140mm Hg, in the proportion of non-smokers and in medication adherence categorized by Morisky scale. In the CHAT-DM study, we assess the proportion of patients achieving HbA1C<7% and in medication adherence as the only two indicators using proportions as secondary outcome. The other secondary outcomes are assessed as a change in mean levels of LDL-C, body mass index (BMI), physical activity and FBG (if necessary), since there is less consensus about the optimal goals for these outcomes, and goals may vary depending on baseline characteristics and clinical conditions. We have clarified the descriptions of outcomes as per below:

Methods and Analysis (page 16, paragraph 2, lines 5-10): “In the CHAT study, the primary outcome is the change in systolic blood pressure (SBP) after 6 months. Secondary outcomes include a change in proportion of patients achieving a SBP<140mm Hg, change in proportion of non-smokers, change in medication adherence categorized by Morisky scale, as well as change in plasma mean level of LDL-C, change in level of body mass index (BMI) and change in level of physical activity.”

Methods and Analysis (page 16, paragraph 3, lines 14-18): “In the CHAT-DM study, the primary outcome is the change in glycemc hemoglobin (HbA1C) as measured by central blood sample. Secondary outcomes include a change in proportion of patients achieving HbA1C<7%, change in medication adherence, as well as change in mean level of FBG, SBP, LDL-C, BMI and physical activity.”

Comment 5: Page 16: line 49: Define the pre-specified subgroups

Response: As suggested by the reviewer, we have edited the description of the pre-specified subgroups as follows:

Methods and Analysis (page 18, paragraph 1, lines 9-12): “We will conduct a pre-specified 2nd analysis with adjustment for patient characteristics as well as subgroup analyses based on age, sex, education, smoking status and tertiles level of endpoints.”

Comment 6: Page 37, line 17: Table 4: “Outcome” is listed as a separate measure, which is too vague.

Response: We have edited accordingly in Table 4, page 40. We added detailed descriptions of outcome in parentheses.

Reviewer #2

Thank you for the opportunity to review this study protocol. I enjoyed reading the manuscript which describes a thoroughly developed mobile health intervention. You have considered the cultural context in which the intervention is rolled out and conducted user testing. It is great to see that mobile health initiatives are becoming part of the research landscape in China-I am looking forward to see the results. I made a number of comments mainly pertaining to language, flow and clarity of the text. Please see my comments below:

Comment 1: Overall: I wonder if you expect a bit too much from the patients?! You target a number of health behaviours and also expect patients to monitor certain things (blood pressure etc). Most people that will enrol in the study will likely have a not so health lifestyle (prior to the study) and there is some research indicating that such people could be overwhelmed by the things they are asked to do. As a result, they might give up early on. Have you considered this?

Response: We recognize that for many people with CHD and DM, it can be overwhelming to make lifestyle changes and simultaneously achieve optimal risk factor control. Yet this is what is expected of patients and is recommended by the AHA/ACCF and Chinese guidelines of secondary prevention for patients with coronary heart disease and diabetes. This study aims to engage patients in achieving these goals in a non-burdensome, non-threatening manner. While prior studies evaluated the effectiveness of mobile phone text messaging to improve single individual health behaviors, few test interventions to improve multiple risk factors, which is a reality for many patients. Moreover, these text messages focusing on multiple behaviors were also acceptable according to the feedback obtained from our patients during user testing and pilot phase. Therefore, we designed this study targeting at multiple health behaviors and intend to explore the efficacy of this kind of intervention.

Comment 2: Overall: I wonder what you mean by semi-personalised text messages?

Response: Here 'semi-personalized text messages' means that some messages are personally addressed with the participant's preferred name. Additionally, we designed the text messages considering a participant's health status. For example, in the CHAT study, non-smoking participants would not receive smoking cessation messages during the study period, and active smokers would receive one smoking cessation message per week. In order to clarify, we added an explanation to methods section and edited the discussion section as follows:

Methods (page 13, paragraph 2, lines 10-12): "The text messages were designed to be semi-personalized with participant's preferred name at beginning of some messages, as well as considering participant's smoking status."

Discussion (page 21, paragraph 3, lines 21-23): "Second, the text messages, though semi-personalized with participant's preferred name and depending on their smoking status, were not tailored specifically to every individual."

Comment 3: Strengths and Limitations (first): I suggest you say "intervention" instead of "solution".

Response: We agree and have modified this word in the manuscript accordingly:

Strengths and Limitations of study (page 4, paragraph 1, lines 2-4): "The main strengths of the study are that it evaluates the efficacy of an innovative, simple, scalable and cost-effective intervention for improving secondary prevention of CHD."

Comment 4: Strengths and Limitations (fourth): I suggest you split up the sentence-after "country".

Response: Thanks for the comment and we have revised the manuscript as follows:

Strengths and Limitations of study (page 4, paragraph 4, lines 16-19): "In contrast to most prior single center text-messaging trials, these two studies are conducted at 37 participating sites spread across a large and geographically diverse country. The results of the studies may be more generalizable."

Comment 5: Introduction: I am not sure what you mean by this sentence: "Prior studies reveal that only one quarter of hospitalized patients do not fill all...". Do you mean that only one quarter obtain medications from their prescriptions? Please make this sentence easier to digest.

Response: We meant that three-fourths of hospitalized patients take all medications from their discharge prescriptions by 120 days after discharge, thus one quarter do not take all medications listed in discharge prescriptions. We modified this sentence as follows:

Introduction (page 5, paragraph 1, lines 7-9): "Prior studies revealed that only three-fourths patients take all medications from their discharge prescriptions by 120 days after discharge."

Comment 6: Introduction: Page 5, line 38: Could you use another word instead of "influential". Maybe say "...are a barrier".

Response: We have revised the sentence as follows;

Introduction (page 5, paragraph 2, lines 16-18): "While high medication costs are a barrier, there is also limited time for education and consultation regarding lifestyle and medication management during clinic visits, which tend to be very brief."

Comment 7: Introduction: Page 5, line-54-55: Could you rewrite this: "Mobile phones already serve as a primary, inexpensive and quick form of communication, alerts and reminders." You could say "Mobile phones are already a primary and inexpensive form of communication, and they are also used to schedule alerts and reminders".

Response: We thank the reviewer for the suggestion and have modified the sentence accordingly:

Introduction (page 5, paragraph 3, lines 22-24): "They are already a primary, inexpensive and quick form of communication, and are also widely used to schedule alerts and reminders."

Comment 8: Introduction: Page 4-5, line-57-7: Could you break up this sentence? I suggest to add the text message information (200,000 messages) to the end of the paragraph as you talk about text messages there.

Response: We have modified the sentence based on the reviewer's suggestions:

Introduction (page 6, paragraph 1, lines 3-7): "Mobile phones are also used across all geographic regions and income levels. Due the ubiquity and convenience of mobile phones, nearly 200,000 mobile phone messages are sent every second in China. Text messaging has the potential to be a scalable and powerful tool to deliver health information."

Comment 9: Introduction: Page 5, line 50-51: Could you restructure the sentence? I think what you mean is that mobile phones, that are ubiquitous in people's lives can be used to deliver interventions that help people to adopt secondary prevention strategies.

Response: We have revised accordingly as follows:

Introduction (page 5, paragraph 3, lines 21-22): "Mobile phones are pervasive and thus can be used to deliver interventions that help people to adopt secondary prevention strategies for CHD in LMICs."

Comment 10: Introduction: page 6, when talking about LMIC studies: Consider adding these 2 very relevant references: Müller, A. M., Alley, S., Schoeppe, S., & Vandelanotte, C. (2016). The effectiveness of e- & mHealth interventions to promote physical activity and healthy diets in developing countries: A systematic review. *International Journal of Behavioral Nutrition and Physical Activity*, 13(1), 109.; Hall, C. S., Fottrell, E., Wilkinson, S., & Byass, P. (2014). Assessing the impact of mHealth interventions in low-and middle-income countries—what has been shown to work? *Global health action*, 7(1), 25606.

Response: We have reviewed these two papers and cited as reference #24 and #29 in the Introduction section on page 6, line 10 and page 6, line 24.

Comment 11: Methods (under Randomization and Blinding): page 8: Delete the extra full stop after last sentence.

Response: Thanks for the comment and we have revised accordingly.

Comment 12: Methods (under trial intervention): page 8, line 51-54: I wonder why you say that the control group received Thank You messages without medical information. I thought the intervention group receives advice on risk factor modification-this doesn't need to mean it is medical (e.g., advice on physical activity is not really medical). So, I suggest you say that the CG did not receive risk factor modification support.

Response: We have revised this sentence as follows:

Methods (page 9, paragraph 2, lines 5-9): "Participants in the intervention groups of the CHAT and CHAT-DM studies receive semi-personalized text messages about CHD risk factor modification for 6 months as well as standard treatment (described below). The control group in both studies receive two thank-you text messages without risk factor modification support each month as well as standard treatment."

Comment 13: Methods (under trial intervention): page 9, line 7-10: Could you restructure this sentence "Participants were also instructed how to withdraw from the study should they desire, by responding with a specific character" to make it easier to digest? You could say "Participants were also informed that they could withdraw from the study by responding to a text message with a specific character."

Response: We have modified the sentence as follows:

Methods (page 9, paragraph 2, lines 16-18): "Participants were also informed that they could withdraw from the study by responding to a text message with a specific character."

Comment 14: Methods (CHAT and CHAT DM intervention development): page 9, last sentence: Could you change to "... to a 140-160 character message in English"?

Response: We have revised this accordingly as follows:

Methods (page 10, paragraph 1, lines 11-13): "All text messages are in Chinese, and each text message is less than 70 Chinese characters, which would be equivalent to a 140-160-character message in English."

Comment 15: Methods (under Phase 1 Approach to...): I just want to let you know that there are two more updated behaviour change technique taxonomies: Michie, S., Ashford, S., Snihotta, F. F., Dombrowski, S. U., Bishop, A., & French, D. P. (2011). A refined taxonomy of behaviour change techniques to help people change their physical activity and healthy eating behaviours: the CALO-RE taxonomy. *Psychology & health*, 26(11), 1479-1498.; Michie, S., Richardson, M., Johnston, M., Abraham, C., Francis, J., Hardeman, W., ... & Wood, C. E. (2013). The behavior change technique taxonomy (v1) of 93 hierarchically clustered techniques: building an international consensus for the reporting of behavior change interventions. *Annals of behavioral medicine*, 46(1), 81-95. This is purely for your information in case you are interested.

Response: These are interesting papers and we appreciate the reviewer informing us of these resources. We have reviewed these two most updated BCT taxonomies and believe that these methods can serve as guidance for evaluating and strengthening our reporting of complex behavioral interventions, thus, we edited the method sections and cite these two papers as references #34 and #35 as follows:

Methods (page 10, paragraph 2, lines 20-22): "In order to strengthening the designing and reporting of theory-based BCT interventions, we took into account the two recent BCT taxonomies by Michie. S, and his colleagues.^{34, 35}"

Comment 16: Methods (under Phase 1 Approach to...): I am very happy that you did this: "Efforts were devoted to selecting the behavioral change techniques most applicable to the Chinese cultural context and compatible with Chinese beliefs and values (Table 1)." You explain what you did and why you did it (considering culture and norms). This is helpful for my own work.

Response: We appreciate this comment.

Comment 17: Methods (Table 1): I found some typos in the table-please correct.

Response: We thank the reviewer for this comment. We have revised the typos.

Comment 18: Methods (Table 1): I generally like the text messages. Just two comments. 1. Sometimes you could give a stronger rationale why a behavior might be good (e.g., message 1 in table). Saying that something is good for health is a bit unspecific and many people might want to know more to also look out for changes (e.g., for physical activity you could say that it might get easier to climb stairs). 2. You sometimes use complicated words in the text messages and I wonder if people will be confused by this? We know that the reading age of the general population is not very high (about 11 years in the UK). However, as you piloted the messages I believe you have checked that. It might also be a language issue here. The messages might be easier to understand in Chinese (for Chinese speakers).

Response: We thank the reviewer for these two comments. Regarding comment 1, we fully agree that providing specific explanations may be better to promote behavior changes. We constructed and used several text messages that are more specific, though they were not described in the article and appendix, for example:

Muscular training is helpful for building the strength of the human body. You can easily find 1-2kg objects to weightlift while engaging in other activities such as watching TV. Common household objects such as cans of food and/or water bottles can be used to exercise, making your goal attainable and flexible for your schedule!

Do you sometimes feel frustrated that you do not have time and energy to exercise after a day's busy work? Try some easier activities in your daily life, e.g. walking, climbing stairs, riding to work and shopping. A short thirty-minute time interval consisting of light and moderate exercise can be also helpful to your health!

Regarding comment 2, we believe this confusion may arise because of the translation. The Chinese messages are easy to understand according to feedback obtained from patients during the pilot phase. We aimed to simplify the messages during development. A linguist reviewed and further refined all 550 text messages, and paid special attention to the cultural meaning of all text messages to ensure better understanding among the targeted population.

Comment 19: Methods (under Phase 2 Expert review): Page 11, line 8-13: I wonder what you mean by "clinical benefit of each text message"? I don't see that a message can have a clinical effect, especially when I look at your outcomes. Can you clarify this in the text?

Response: We agree and have modified the sentence, substituting 'clinical benefit' with 'practical usefulness' on page 11, line 20.

Comment 20: Methods (under Phase 3 User testing and pilot study): I don't understand how the numbers for the user testing add up. You say you checked 19 text messages with 39 people ($19 * 39=741$). However, you also indicate that 630 was the total. Would be great if you could say what was actually calculated and how.

Response: There were two versions of messages used for testing with 39 patients, one was for smokers and the other one was for non-smokers. We checked 17 messages with 14 smokers and 16 messages with 25 non-smokers; the missing were 8, so in total the number was 630.

Comment 21: Methods (under data collection): Page 14, line 21-25: Please rewrite this sentence to make it easier to understand.

Response: Thanks for the comment and we have revised this sentence as follows:

Original sentence:

Methods (page 14, paragraph 1, lines 21-25): 'Follow-up information is conducted at 6 months by personal interviewing with research staff again collecting the information above (Table 4).'

After revision:

Methods (page 15, paragraph 1, lines 6-7): "Research staff would collect the above information again as listed in Table 4 at follow-up visit."

Comment 22: Methods (Table 4): What does the variable "Outcome" mean (there are ticks for it).

Response: We have modified Table 4, on page 40, accordingly. We added detailed descriptions of outcome in parentheses.

Comment 23: Methods (under data management): Page 14, line 38-43: Please rewrite this sentence to make it easier to understand (esp., last part).

Response: As suggested, we have modified this paragraph as follows:

Methods and Analysis (page 15, paragraph 2, lines 16-18): "Additionally, this web-based platform is used to monitor project progress, as well as provide management support for hospitals, staff members, equipment, sampling collection and transfer."

Comment 24: Methods (under outcomes): Page 15, line 14: I suggest to add a "the" after "In" (In the CHAT study). Same for the CHAT DM study.

Response: We have modified the sentence accordingly.

Comment 25: Methods (under statistical analysis): I wonder why you measure a number of variables but do not plan to control for them (e.g., age). Also, I suggest to use an ANCOVA on the change scores (baseline as one covariate).

Response: We appreciate this comment. Although by the nature of our study's design, there would not be a difference in patient characteristics between two groups. Nevertheless, giving the sample size in our study, some patient characteristics might show a marked difference between the groups. To address this potential challenge, we will conduct a 2nd analysis with adjusting for patient characteristics. In addition, the aim of collecting variables including age is to achieve a balance of participants' characteristics in both arms using a stratified randomization approach. We may also use this variable during a subsequent sub-group analysis.

Additionally, we have revised the statistical methods as the reviewer recommended:

Methods and Analysis (page 18, paragraph 2, lines 6-8): "The primary analysis will employ analysis of covariance (ANCOVA) with baseline values of the analyzed endpoints used as covariates when appropriate."

Comment 26: Methods (under statistical analysis): Page 16, line 49-51: Please reword this sentence as it seems to not be correct.

Response: We have edited the sentence as follows:

Methods and Analysis (page 18, paragraph 1, lines 9-12): "We will conduct a pre-specified 2nd analysis with adjustment for patient characteristics as well as subgroup analyses based on age, sex, education, smoking status and tertiles level of endpoints."

Comment 27: Methods (under statistical analysis): Why do you think the SBP reduction in the intervention group will be smaller?? Isn't the goal of the intervention to reduce this more than in the control group?

Response: We apologize for the confusing description. We meant that "considering the compliance to the intervention, the SBP reduction could be smaller than expected effect of our intervention", rather than that "the SBP reduction in the intervention group will be smaller than in the control group". To avoid this potential confusion, we have modified the description accordingly as follows:

Methods and Analysis (page 18, paragraph 2, lines 17-24; page 19, paragraph 1, lines 1-5): "All sample size calculations in both studies are for 80% power and 0.05 (one-sided) level of significance, allowing for 20% dropout rate during follow up. In the CHAT study, we estimate a mean SBP level of 132mm Hg [SD, 18mm Hg] in the study population according to the report of previous studies, and assume an absolute reduction in SBP of 5 mm Hg at 6 months from baseline. A total sample size of 800 patients would be adequate to detect this difference in SBP between two groups even when considering the potential compliance to the intervention. In the CHAT-DM study, we assume a mean HbA1C level of 7.2% [SD, 1.6%] based on data from studies involving similar populations and supposed a 0.5% absolute decrease in HbA1C across treatment group. A total sample size of 500 patients would be adequate to detect this difference in HbA1C between two groups even when considering the potential compliance to the intervention."

Comment 28: Methods (under statistical analysis): I am not sure I understand your sample size calculation correctly. I assume you want to have a difference in change between the 2 groups of 3.5mm HG? Is that correct?

Response: We apologize for any confusion. We modified the description as cited in Comment 27.

Comment 29: Discussion: Page 18, line 10: I suggest you use "intervention" instead of "solution".

Response: We agree and have modified this word in the manuscript accordingly:

Discussion (page 19, paragraph 3, lines 16-19): "The CHAT and CHAT-DM studies aim to assess the efficacy of an innovative intervention for improving secondary prevention of CHD by using simple and cost-effective text-messaging technology among patients with known CHD, with or without DM, throughout China."

Comment 30: Discussion: Page 18, line 32-41: You make an argument for interventions that target multiple risk factors. As I indicated earlier, I am not sure that targeting multiple risk factors at once is a good idea as people might be overwhelmed. In your argument, I don't understand why you think this approach is patient-centred (please correct from "centric" to "centred")?? The references you cite are fairly old-in this study promoting many health behaviours did not work: Reinwand, D. A., Schulz, D. N., Crutzen, R., Kremers, S. P., & de Vries, H. (2015). Who follows eHealth interventions as recommended? a study of participants' personal characteristics from the experimental arm of a randomized controlled trial. *Journal of medical Internet research*, 17(5). You could rephrase your Discussion slightly to give a more balanced view.

Response: We target multiple risk factors, as opposed to just one, as we believe that this approach is more consistent with the challenges that people with CHD and DM face. Patients with CHD and DM often have multiple risk factors, yet the care for these conditions can be fragmented. Cardiovascular health is multidimensional, and health messages that target numerous risk factors may help in promoting a model of health, and in illuminating connections between risk factors. In this way, we believe we are testing a more patient-centered rather than disease-centered approach to cardiovascular health. While the Australian study TEXT-ME found that text messages that targeted multiple risk factors was effective, it is unknown whether this approach will also be effective in China. The study by Reinwand tested an email intervention, and it is not clear whether email messages versus text messages are equally effective. Accordingly, we modified the discussion section as follows:

Discussion (page 20, paragraph 1, lines 1-10): "The CHAT and CHAT-DM studies have several strengths. While prior studies evaluated the effectiveness of mobile phone text messaging to improve single individual health behaviors, very few trials have addressed the role of text messages in managing multiple risk factors in patients with CHD or DM, which is a reality for many patients. Patients with CHD and DM often have multiple risk factors, yet the care for these conditions is often fragmented. Targeting multiple risk factors concurrently instead of managing a single risk factor may be more efficient and impactful on disease management as this strategy is more patient-centered than disease-centered, and have a greater likelihood of improving risk factor control and cardiovascular outcomes."

Comment 31: Discussion: Page 19, line 32-39: I suggest you split up this sentence. The part "...the results" could be a new sentence to make your point more prominent.

Response: We have modified the sentence as recommended:

Discussion (page 21, paragraph 2, lines 6-9): "In contrast to most prior single center text-messaging trials, these two studies are conducted at 37 participating sites spread across a large and geographically diverse country. These results may be more generalizable."

Comment 32: Discussion: Page 20, line 5-7: After "validated" add "are" (widely used), and say "these metrics".

Response: We have modified the sentence as follows:

Original sentence:

Discussion (page 20, paragraph 1, lines 3-7): "However, we believe that any such bias would be balanced across the treatment and control groups, and MMAS-8 and IPAQ have been validated and widely used in measuring this metric."

After revision:

Discussion (page 21, paragraph 3, lines 18-20): "However, we believe that any such bias would be balanced across the treatment and control groups, and MMAS-8 and IPAQ have been validated and are widely used in measuring these metrics."

Comment 33: Discussion: Page 20, line 8-12: You say here that the messages were not tailored, but earlier you said they were. Please be consistent. For me, it seems as if the messages were generic/not tailored.

Response: The messages were not tailored for each individual, though they were semi-personalized depending on smoking status. We have clarified this language as follows:

Discussion (page 21, paragraph 3, line 21-23): "Second, the text messages, though semi-personalized with participant's preferred name and depending on their smoking status, were not tailored specifically to every individual, which may reduce their efficacy."

Comment 34: Discussion: Page 20, line 37-44: Consider rewording this sentence to make it easier to understand.

Response: We appreciate this comment and have reworded the sentence as follows:

Original sentence:

Discussion (page 20, paragraph 2, lines 37-44): 'If this innovative and simple prevention were proven to help, considering the low marginal cost and anticipated minimal adverse events, even with very modest effects, its benefits would be substantial in a country as large and populous as China.'

After revision:

Discussion (page 22, paragraph 1, lines 10-13): "Considering that text messages are low in cost and incur minimal, if any, risk, if this innovative and simple prevention were proven to be helpful, its benefits would be substantial in a country as large and populous as China."

Comment 35: Discussion: Page 20, line 53: You say the RCT will check the efficacy of text messages in LMICs. Please reconsider this statement: You only run the study in China. You cannot generalise to other LMICs (especially, those that are very different compared to China).

Response: We have substituted 'LMICs' with 'China' accordingly:

Discussion (page 22, paragraph 2, lines 14-17): "In conclusion, the CHAT and CHAT-DM studies are multi-center, randomized controlled trials that are being conducted at 37 hospitals, and will thoroughly test the efficacy of text-messages to support secondary prevention for CHD and DM in China."

Comment 36: Discussion: last sentence: 1-You use past tense here (targeted), but the study is still under way. So, you could say "targets". 2-break up this sentence to make it easier to understand. I suggest a full stop after "intervention". 3-may just say that the intervention can potentially benefit a significant portion of the population (instead of "over thousand").

Response: We thank the reviewer for this comment, and have modified the manuscript in response to the above points as follows:

Original sentence:

Discussion (page 20, paragraph 3, lines 47-58): “The studies targeted high-risk patient populations with a culturally-sensitive, scalable and cost-effective text-message intervention, went through comprehensive data collection and rigorous data management, and have the potential to provide novel insights into disease management and be scaled-up to improve health in over a thousand patients in future.”

After revision:

Discussion (page 22, paragraph 2, lines 17-22): “The studies target high-risk patient populations with a culturally-sensitive, scalable and cost-effective text-message intervention. These two trials go through comprehensive data collection and rigorous data management, and have the potential to provide novel insights into disease management and be scaled-up to improve health in a significant proportion of patients in future.”

VERSION 2 – REVIEW

REVIEWER	Lijing Yan Duke Kunshan University
REVIEW RETURNED	02-Oct-2017

GENERAL COMMENTS	The authors adequately responded to all reviewers' comments. This reviewer does not have more comments. Suggest accept.
---

REVIEWER	Andre Matthias Müller Saw Swee Hock School of Public Health, National University of Singapore, Singapore
REVIEW RETURNED	21-Sep-2017

GENERAL COMMENTS	The authors of this interesting manuscript did a great job revising the manuscript. There are just a few (mainly) language issues I want to point out. Please see my comments below: Abstract (page 2, line 15): Could you say “text-message based” as this makes it clear that you use a technology and not only text (like printed text)? Introduction (page 5, paragraph 1, lines 7-9): Please add "of all" where indicated: “Prior studies revealed that only three-fourths of all patients take all medications from their discharge prescriptions by 120 days after discharge.” Introduction (page 6, line 1): Please remove “already”. Introduction (page 6, paragraph 1, lines 3-7): Please add "As such" where indicated: “Mobile phones are also used across all geographic regions and income levels. Due the ubiquity and convenience of mobile phones, nearly 200,000 mobile phone messages are sent every second in China. As such, text messaging has the potential to be a scalable and powerful tool to deliver health information.” Introduction (page 6, line 23): Please remove “generalizability”. Methods (page 8, line 8): Can you say “have” instead of “having”? Methods (page 8, lines 14 to 16): I am not sure this sentence is correct. It doesn't seem right. You say you maintained a log of people you are not eligible but declined. If they are not eligible they should not be able to decline.
--

	Please correct this. Methods (page 13, paragraph 2, lines 10-12): I appreciate you revised this but I think the language needs to be improved. My suggestion: "The text messages were semi-personalized with participant's preferred name at the beginning of some messages, and the smoking status being the second factor considered for personalisation. Methods (page 10, paragraph 2, lines 20-22): Please remove "his" after "Michie. S. and" as Susan Michie is a woman. I would just say "Michie and colleagues". Methods (under statistical analysis): Just a comment that ANCOVA cannot adjust for the difference in variables between groups. It is a technique that helps to get a cleaner effect by adjusting for variables that are likely to impact the outcome. If you have differences between groups in, for example, age these cannot be controlled for with ANCOVA-the differences in the groups are there and there is nothing you can do about it. So, adjusting for patient characteristics can be done in a second step (or initially) but it cannot reduce the flaws of having imbalanced groups. Methods (page 18, lines 19 and 20): I think this sentence is a bit unclear. Maybe say "The mean change of each risk factor will also be compared between groups." And then after the next sentence you could just say "In all analysis 95% Confidence Intervals and p-values will be reported". Make the changes as appropriate. Methods and Analysis (page 18, paragraph 2, lines 17-24; page 19, paragraph 1, lines 1-5): For your sample size calculation I think what you want is a difference in change of outcomes between groups and not only a reduction of the outcome from baseline to 6 months. I suggest to add this information as otherwise I don't see a point of having a control group (you want to see if the IG changes more then the CG). I also suggest that you say "non-compliance" for both studies. Discussion (page 20, paragraph 2, lines 37-44): This sentence is hard to understand. My suggestion: Text-messaging interventions are simple, cost-effective and incur minimal, if any, risk. Hence, if our interventions are effective in reducing/improving XYZ [insert outcomes] large-scale implementation could benefit a wide population of China. Discussion (page 21, paragraph 3, lines 21-23): I appreciate you revised this but I think the language needs to be improved. My suggestion: "Second, the text messages, though semi-personalized (participant's preferred name, smoking status) were not tailored specifically to every individual." Discussion (page 22, paragraph 2, lines 15-17): I would remove this sentence as this is not necessary to say. Discussion (page 22, paragraph 2, lines 18-22): Add "can" before "be scaled-up". Add "the" before "future". References: I wonder why you use "etc"? I have never seen this and think "et al." is the convention. This is also what BMJopen uses as far as I know.
--	---

VERSION 2 – AUTHOR RESPONSE

Reviewer #1

The authors adequately responded to all reviewers' comments. This reviewer does not have more comments. Suggest accept.

Response: Thank you.

Reviewer #2

The authors of this interesting manuscript did a great job revising the manuscript. There are just a few (mainly) language issues I want to point out. Please see my comments below:

Comment 1: Abstract (page 2, line 15): Could you say “text-message based” as this makes it clear that you use a technology and not only text (like printed text)?

Response: We have substitute “text-based” with “text-message based” accordingly.

Comment 2: Introduction (page 5, paragraph 1, lines 7-9): Please add "of all" where indicated: “Prior studies revealed that only three-fourths of all patients take all medications from their discharge prescriptions by 120 days after discharge.”

Response: We thank the reviewer for the suggestion and have revised the sentence as follows: Introduction (page 5, paragraph 1, lines 7-9): “Prior studies revealed that only three-fourths of all hospitalized patients take all medications from their discharge prescriptions by 120 days after discharge.”

Comment 3: Introduction (page 6, line 1): Please remove “already”.

Response: We have removed this word from the sentence accordingly.

Comment 4: Introduction (page 6, paragraph 1, lines 3-7): Please add "As such" where indicated: “Mobile phones are also used across all geographic regions and income levels. Due the ubiquity and convenience of mobile phones, nearly 200,000 mobile phone messages are sent every second in China. As such, text messaging has the potential to be a scalable and powerful tool to deliver health information.”

Response: We have revised the sentence as suggested: Introduction (page 6, paragraph 1, lines 4-7): “Due to the ubiquity and convenience, nearly 200,000 mobile phone messages are sent every second in China. As such, text messaging has the potential to be a scalable and powerful tool to deliver health information.”

Comment 5: Introduction (page 6, line 23): Please remove “generalizability”.

Response: We have removed this word from the sentence accordingly.

Comment 6: Methods (page 8, line 8): Can you say “have” instead of “having”?

Response: We have modified the sentence accordingly as follows:

Methods (page 8, paragraph 2, lines 4-6): "In CHAT, patients were eligible if they had established CHD defined as a history of AMI and/or percutaneous coronary intervention (PCI), access to a mobile phone to read and send text messages, and did not have diabetes."

Comment 7: Methods (page 8, lines 14 to 16): I am not sure this sentence is correct. It doesn't seem right. You say you maintained a log of people you are not eligible but declined. If they are not eligible they should not be able to decline. Please correct this.

Response: We have modified the sentence as follows:

Methods (page 8, paragraph 2, lines 11-13): "A 'screening log' of basic demographic information and reasons for not participating in patients deemed ineligible or declined to participate has been recorded."

Comment 8: Methods (page 13, paragraph 2, lines 10-12): I appreciate you revised this but I think the language needs to be improved. My suggestion: "The text messages were semi-personalized with participant's preferred name at the beginning of some messages, and the smoking status being the second factor considered for personalisation."

Response: We have revised this sentence according to the reviewer's suggestion as follows:

Methods (page 13, paragraph 2, lines 12-14): "The text messages were semi-personalized with participants' preferred name at the beginning of some messages, and the smoking status being the second factor considered for personalization."

Comment 9: Methods (page 10, paragraph 2, lines 20-22): Please remove "his" after "Michie. S. and" as Susan Michie is a woman. I would just say "Michie and colleagues".

Response: We have revised the sentence as follows:

Methods (page 10, paragraph 2, lines 21-23): "In order to strengthen the designing and reporting of theory-based BCT interventions, we took into account the two recent BCT taxonomies by Michie and colleagues."

Comment 10: Methods (under statistical analysis): Just a comment that ANCOVA cannot adjust for the difference in variables between groups. It is a technique that helps to get a cleaner effect by adjusting for variables that are likely to impact the outcome. If you have differences between groups in, for example, age these cannot be controlled for with ANCOVA-the differences in the groups are there and there is nothing you can do about it. So, adjusting for patient characteristics can be done in a second step (or initially) but it cannot reduce the flaws of having imbalanced groups.

Response: We appreciate the reviewer for this comment and agree with the observation about the problems with imbalanced groups.

Comment 11: Methods (page 18, lines 19 and 20): I think this sentence is a bit unclear. Maybe say "The mean change of each risk factor will also be compared between groups." And then after the next sentence you could just say "In all analysis 95% Confidence Intervals and p-values will be reported". Make the changes as appropriate.

Response: We have modified accordingly as follows:

Methods and Analysis (page 18, paragraph 1, lines 10-12): "The mean change of each risk factor will also be compared between groups. In all analyses, 95% Confidence Intervals and two-sided p values will be reported."

Comment 12: Methods and Analysis (page 18, paragraph 2, lines 17-24; page 19, paragraph 1, lines 1-5): For your sample size calculation I think what you want is a difference in change of outcomes between groups and not only a reduction of the outcome from baseline to 6 months. I suggest to add this information as otherwise I don't see a point of having a control group (you want to see if the IG changes more than the CG). I also suggest that you say "non-compliance" for both studies.

Response: We have modified the section as follows:

Methods and Analysis (page 18, paragraph 2, lines 15-24; page 19, paragraph 1, lines 1-5): "All sample size calculations in both studies are for 80% power and 0.05 (one-sided) level of significance, allowing for 20% dropout rate during follow up. In the CHAT study, we estimate a mean SBP level of 132mm Hg [SD, 18mm Hg] in the study population according to the report of previous studies, and assume an absolute difference in SBP changes of 5 mm Hg at 6 months from baseline in the intervention group and no change in the control group. A total sample size of 800 patients would be adequate to detect this difference in SBP changes between two groups even when considering the potential non-compliance to the intervention. In the CHAT-DM study, we assume a mean HbA1C level of 7.2% [SD, 1.6%] based on data from studies involving similar populations and supposed a 0.5% absolute difference in HbA1C changes in treatment group and no change in the control group at 6 months from the baseline. A total sample size of 500 patients is adequate to detect this difference in HbA1C change between two groups even when considering the potential non-compliance to the intervention."

Comment 13: Discussion (page 20, paragraph 2, lines 37-44): This sentence is hard to understand. My suggestion: Text-messaging interventions are simple, cost-effective and incur minimal, if any, risk. Hence, if our interventions are effective in reducing/improving XYZ [insert outcomes] large-scale implementation could benefit a wide population of China.

Response: We have edited the sentence as follows:

Discussion (page 22, paragraph 1, lines 12-15): "Text-messaging interventions are simple, cost-effective and incur minimal, if any, risk. Hence, if our interventions are effective in improving risk factor control and lifestyle modification, large-scale implementation could benefit a diverse population in China."

Comment 14: Discussion (page 21, paragraph 3, lines 21-23): I appreciate you revised this but I think the language needs to be improved. My suggestion: "Second, the text messages, though semi-personalized (participant's preferred name, smoking status) were not tailored specifically to every individual."

Response: We have modified the sentence accordingly:

Discussion (page 21, paragraph 3, lines 24-25): "Second, the text messages, though semi-personalized (participant's preferred name, smoking status) were not tailored specifically to every individual."

Comment 15: Discussion (page 22, paragraph 2, lines 15-17): I would remove this sentence as this is not necessary to say.

Response: We have removed this sentence.

Comment 16: Discussion (page 22, paragraph 2, lines 18-22): Add "can" before "be scaled-up". Add "the" before "future".

Response: We have modified the sentence as suggested:

Discussion (page 22, paragraph 2, lines 17-20): "These two trials target high-risk patient populations with a culturally-sensitive, scalable and cost-effective intervention, and have the potential to provide novel insights into disease management and can be scaled-up to improve health in a significant proportion of patients in the future."

Comment 17: References: I wonder why you use "etc"? I have never seen this and think "et al." is the convention. This is also what BMJopen uses as far as I know.

Response: We have reformatted the references in accordance with the style of BMJ Open.

VERSION 3 – REVIEW

REVIEWER	Andre Matthias Müller Saw Swee Hock School of Public Health, National University of Singapore, Singapore
REVIEW RETURNED	27-Oct-2017
GENERAL COMMENTS	Thank you for revising the manuscript appropriately. I have no more comments.